# Technology Transfer Centers as Support Instruments for SMEs—Comparative Analysis of Poland and Malaysia

Maciej Woźniak [1,*]🄳, Marek Matejun [2]🄳, Fadhlur Rahim Azmi [3]🄳, Mior Harris Mior Harun [4] and Fazlena Hamzah [5]

1 Faculty of Management, AGH University, 30-067 Cracow, Poland
2 Faculty of Management, University of Lodz, 90-237 Lodz, Poland; marek.matejun@uni.lodz.pl
3 Faculty of Business & Management, Universiti Teknologi MARA, Cawangan Melaka Kampus Bandaraya Melaka, 110, Off Jalan Hang Tuah, Melaka 75350, Malaysia; fadhlur@uitm.edu.my
4 Arshad Ayub Graduate Business School, Universiti Teknologi MARA, Arau 02600, Malaysia; miorharris@uitm.edu.my
5 Business Innovation & Technology Commercialization Centre (BITCOM), Universiti Teknologi MARA, Arau 02600, Malaysia; fazlena@uitm.edu.my
* Correspondence: mawozniak@agh.edu.pl; Tel.: +48-6174347

**Abstract:** The goal of the paper is to compare technology transfer centers in Poland and Malaysia. Therefore, the authors decided to use the comparative analysis method. The findings show that technology transfer and commercialization efforts both in Poland and Malaysia are on the right track. This demonstrates the universities' persistent dedication to turning research and innovative ideas into concrete products, as seen by the university's sustained growth in total product commercialization. It emphasizes the critical role that they play in promoting technological transfers, particularly for SMEs. The paper contributes to the macroeconomics theory in the area of public policy. Furthermore, it also provides insights into the theory of incentives, particularly in the field of non-financial support. The findings could be of interest to policymakers on macro and micro levels.

**Keywords:** technology transfer centers (TTCs); small and medium-sized companies (SMEs); support for enterprises

## 1. Introduction

Technology transfer plays a pivotal role in fostering economic growth, innovation, and competitiveness in today's globalized world. It is a process that involves the exchange of knowledge, expertise, and technology between various stakeholders, such as government agencies, academia, research institutions, and industry players. This transfer of technology can occur both domestically and internationally, leading to significant advancements in various sectors. Therefore, it is a vital part of the Industry 4.0 revolution and leads to the digitalization of the economy. It also helps to achieve the main goals of sustainable development.

Poland, a European Union member with a diverse and growing economy, has experienced its own unique journey in technology transfer. As a country transitioning from a centrally planned to a market-oriented economy, Poland has faced the challenge of integrating advanced technologies into its industries and harnessing innovation for economic growth [1]. Poland's technology transfer landscape involves collaborations between universities, research institutions, and industry partners, often facilitated by government initiatives and European Union funding programs. These efforts have aimed to bridge the gap between research and commercialization, fostering innovation-driven entrepreneurship.

On the other hand, Malaysia, a rapidly developing nation in Southeast Asia, has placed a strong emphasis on technology transfer as a means to accelerate industrial growth and enhance its global competitiveness. The Malaysian government has implemented various initiatives and policies to facilitate the exchange of technology and knowledge

across different sectors. These efforts aim to leverage external expertise and resources to support domestic industries and drive economic transformation. One of the notable programs in Malaysia is the offset program known as the Industrial Collaboration Program, which was designed to facilitate collaboration between local agencies and international partners, ensuring "value for money" in procurements [2]. Additionally, collaborative relationships with multinational corporations have helped elevate the quality of regional products and services while enhancing cost-effectiveness [3]. The effects of training on productivity among local businesses and professionals have also been a focus of technology transfer efforts [4].

Technology transfer has been given top priority in Malaysia in an effort to boost its industrial growth and international competitiveness. To encourage the transfer of technology and information among diverse industries, the Malaysian government has put in place a number of initiatives and policies. In addition to improving cost-effectiveness, cooperative partnerships with global firms have raised the caliber of local goods and services [5].

In connection with the above, the goal of the paper is to compare technology transfer centers (TTCs) in Poland and Malaysia. By examining these two distinct countries on similar levels of development, we can gain valuable insights into the role of technology transfer in driving economic development and innovation in diverse national contexts. In order to achieve this goal, the authors decided to use the comparative analysis method.

The paper contributes to the macroeconomics theory in the area of public policy. Furthermore, it also provides insight into the theory of incentives, particularly in the field of non-financial support. The paper is organized as follows. In the second section, the literature review is performed. Section 3 shows the data collected and describes the methodology. Section 4 presents research results related to technology transfer centers in Poland and Malaysia. Section 5 provides a discussion of the results obtained in the context of previous studies, a literature review as of relevant and promising research directions. The conclusions bring together the most important findings of the study, explain the constraints of the paper, and give directions for future research.

## 2. Literature Review

### 2.1. Poland

Poland, as a member of the European Union (EU), has to follow its regulations. In order to achieve the sustainable development goals, Poland has to follow, among others, the directive of the European Parliament on the promotion of the use of energy from renewable sources. It assumes that a minimum of 27% of the energy consumed in the EU is supposed to come from renewable sources by 2030 [6]. However, Poland is amongst the countries of the European Union with the highest share of fossil fuel production. This decade will be very important in the process of decarbonizing the electrical system of Europe in order to ensure sustainable growth. Poland faces a dramatic challenge as the producer of the dirtiest electricity in the EU. The fastest way to achieve significant reductions in fossil fuel consumption is to develop solar and wind energy. Therefore, there is a need to develop new technologies and transfer them to companies. The experts state that the most important task is to develop onshore wind farms. Developing only offshore is not enough to achieve this goal. This process requires the transfer of new technology from the science institution to companies.

However, companies in Poland prefer to invest in research and development (R&D) activities rather than acquire knowledge from external entities. As far as industrial enterprises are concerned, only a few of them (2–3%) used inventive projects of domestic external partners that are protected by exclusive rights. In connection to this, Jarzynowski [7] states that intellectual property rights (IPR) in Poland is in the early stage of development. Therefore, the number of commercialized technologies in the previous decade varied from 8 to 21 each year. The result was quite poor. In connection to this, some support schemes were introduced. Open Innovation Network offers grants for small and medium-sized

enterprises (SMEs) that want to purchase, among other patents, industrial designs, or IPR. Moreover, the companies are supported by technology brokers on the stage of preparing an application. Furthermore, a special platform has been created to connect enterprises with technology providers. Their transactions can be supported by financial aid.

*2.2. Malaysia*

Malaysia intends on strong economic growth and the continuous expansion of real per capita income. In developing the country, the government, from time to time, conducts large procurements, mainly for the construction of megaprojects. Thus, Malaysia leverages many programs to achieve its goals by the imports of technology rather than procuring it locally. This option is chosen due to the unavailability of the technology locally, or it is the best option to speed up the development of the project to meet the time and objectives of the procurement [8]. Malaysia needs to be aggressive in moving forward, particularly in the area of technology development, to stay competitive in the global economy. The New Economic Model introduced aims to create a high-income society with sustainability and inclusiveness, leading to a high quality of life. Therefore, Malaysia needs to develop further the required capability and capacity, in particular related to technology, as a catalyst to leapfrog the nation to become a knowledge-based economy [2].

According to Awang et al. [4], however, the private sector in Malaysia lags behind the public sector in terms of research and technical advancement. Consequently, Malaysia has used technology transfer as a method to transition from a developing nation to a developed one. Malaysia's participation in high-value-added activities has been facilitated by the transfer of technology, particularly due to the lack of local skills in areas such as service bargaining, local design and engineering. Several national programs, including the automotive industry development [9], the rail industry advancement [10,11], the agro-based industry initiatives [4,12], and an offset program utilizes the Industrial Collaboration Program (ICP), have utilized technology transfer to drive progress [2].

Conforming to the governments "value for money" procurement policy, the technology transfer program in Malaysia works to fulfill the procurement needs of government agencies [2]. Therefore, technological transfers entail some economic interaction between recipients and providers, which is frequently a prerequisite for the selling of products and services.

The primary objective of technology transfer in Malaysia is to promote industrial growth through locally controlled firms. Collaborative ties with international firms improve regional product quality and enable cost-effective service delivery [3]. Training activities also contribute to enhanced productivity among smallholders [4], while good management methods play a crucial role in knowledge transfer to local professionals.

*2.3. Poland and Malaysia*

The distinct cultural and social backgrounds of Poland and Malaysia, coupled with their geographical separation, impact the technology transfer process. Furthermore, the kinds of technologies that are transmitted and implemented depend on the unique energy demands and goals of each nation. Poland relies on fossil fuels, whereas Malaysia prioritizes renewable energy sources like wind and solar [13]. Due to the necessity of adapting innovations to the local context, these variations offer opportunities and problems for technology transfer [13].

However, both Poland and Malaysia are fast-developing economies. The value of Gross Domestic Product (GDP) per capita based on Purchasing Power Parity (PPP) in these countries 25 years ago was at a similar and low level, around 14.6 thousand USD. Since then, they made spectacular progress. The growth rate was, however, a little faster in Poland. Therefore, the value of GDP per capita PPP has risen to about 36.7 thousand USD in the Polish case, whereas it has increased to around 28.3 thousand USD in the Malaysian case.

In both countries, public support is particularly important for SMEs as most of them have resource limitations. One of the solutions is to leverage external resources by coop-

eration with technology transfer centers [14]. These organizations have a significant role in promoting and liaising between the parties [6]. As a result, transferring technology to local workforces is an essential first step in gaining a sustained competitive edge. The government should, nevertheless, enhance its current technology transfer strategy by emphasizing innovation as one of the key factors for a successful technology transfer outcome, digitizing the technology transfer process, and creating a technology transfer office within a government strategic procurement project [11]. Unfortunately, the research regarding technology transfer is focused only on one country: Poland or Malaysia. There is a lack of comparative analysis on an international level. In connection with this, the following research question emerged:

How do TTCs in Poland and Malaysia support technology transfer from science institutions to SMEs?

The answer to this question requires to use of the comparative analysis method.

## 3. Materials and Methods

### 3.1. Materials

Comparing technology transfer centers (TTCs) in Poland and Malaysia with an emphasis on business sectors, technological advancements, and common ground challenges was our goal. The regulatory environment in both nations, including the laws and incentives that encourage knowledge transfer, was investigated in order to find common ground issues for technology transfer [15]. Collaboration and communication styles are two examples of cultural and social elements that affect the technology transfer process that was taken into account. An evaluation was conducted on the factors that affect technology transfer between the two countries, including industry-specific characteristics, market conditions, and economic position. Furthermore, the disparities between Poland's and Malaysia's energy priorities and conditions were examined, with a special emphasis on the latter's renewable energy programs and obstacles [15].

The study looked at the business sectors and technological innovations that are most active in technology transfer, such as manufacturing, IT, and engineering. Given the increased emphasis on renewable energy worldwide, the transfer of cutting-edge solar and wind energy technologies between the two nations was also investigated [16]. Innovations in medical technology and the healthcare sector were taken into consideration, as they frequently have a significant impact on technology transfer. Cooperation between academic institutions and business leaders in management, finance, and education was also investigated.

In this study, data were collected between September and October 2023 from the following sources:

- governmental reports,
- academic publications,
- institutional records.

In the case of the industrial records, we used two different sources of data. The first one was an official database about the TTCs in Poland and Malaysia. However, access to the second source about conducted technology transfers from universities to SMEs was much more difficult. Some data are top secret. Therefore, we only obtained information about the cumulative numbers of transfers the universities, TTCs to small and medium-sized enterprises. Our sample consists of 200 commercialization's together.

### 3.2. Methods

We used desk research and then comparative analysis in the paper. Desk research is also known as secondary research. This is a method that involves the collection and analysis of existing information and data from published sources. This type of research is conducted without direct involvement in data collection, such as surveys or experiments. Instead, we gather and review information that has already been produced by others. Desk research is commonly used in various fields and is useful research on a wide range of topics.

It helps researchers build on existing knowledge, identify gaps in the literature, and inform decision-making processes. However, it may have limitations, such as potential biases in the existing data and the inability to gather specific or tailored information. Therefore, desk research is often complemented by primary research methods when a more in-depth and specific investigation is required.

A comparative analysis is a method also used in various fields to evaluate two or more entities or subjects to identify similarities, differences, strengths, weaknesses, and patterns. This type of analysis is designed to gain a deeper understanding of the subjects being compared. When conducting a comparative analysis, researchers typically follow a structured process that includes collecting relevant data and using various analytical methods, such as statistical analysis or qualitative assessment. The goal is to identify patterns or gain insights that may not be apparent when examining subjects in isolation. Comparative analysis is a valuable tool for problem-solving and academic research, as it allows for a deeper understanding of the subject matter through a systematic and structured examination of similarities and differences.

Moreover, we calculated the dynamics of technology transfer change in Poland and Malaysia.

## 4. Results

There was a dynamic growth in the number of technology transfer centers in Poland in the second half of the 90ties of the previous millennium and the first decade of the present century. Then, it began to decrease significantly. Their number has started again to increase but slowly since 2015 [17]. Nowadays, there are over 60 CTTs in Poland. They are presented in Table 1.

**Table 1.** Transfer technology centers in Poland.

| Name of the Transfer Technology Centre | Organization |
| --- | --- |
| All universities and science institutions | |
| AERO-PRz and CTT PRz | Rzeszow University of Technology |
| Bio & Technology Innovations Platform | Technology Transfer Center BioTech-IP International Institute of Molecular and Cell Biology in Warsaw |
| Office of Cooperation with the Economy | University of Silesia in Katowice |
| Center for Information Technology and Transport Safety | University of Economics and Innovation in Lublin |
| The Innovation Center of the Maritime University of Szczecin | Maritime University of Szczecin |
| Center for Innovation and Research Commercialization | Maria Curie Sklodowska University |
| Center for Innovation and Technology Transfer | Wroclaw Medical University |
| Center for Innovation and Technology Transfer | University of Warmia and Mazury in Olsztyn |
| Center for Innovation and Technology Transfer of the Silesian University of Technology | Silesian University of Technology |
| Center for Innovation and Technology Transfer of Warsaw University of Life Sciences | Warsaw University of Life Sciences |
| Center for Innovation and Technology Transfer of the Medical University of Lodz | Medical University of Lodz |
| Center for Innovation and Knowledge Transfer at the University of Wroclaw | University of Wroclaw |
| Center for Innovation, Development and Technology Transfer of the Poznań University of Technology | Poznan University of Technology |

**Table 1.** *Cont.*

| Name of the Transfer Technology Centre | Organization |
|---|---|
| Center for Innovation, Technology Transfer and Development of CITTRU University | Jagiellonian University |
| Center for Entrepreneurship and Technology Transfer (CPTT) | University of Zielona Gora |
| Technology Transfer Center | The State University of Applied Sciences in Elbląg |
| Technology Transfer Center | Pomeranian Medical University in Szczecin |
| Technology Transfer Center | University of Lodz |
| Technology Transfer Center | Cracow University of Technology |
| Center for Technology Transfer of AGH | AGH University of Science and Technology |
| Center for Transfer of Marine Technology | Maritime University of Szczecin |
| Technology Transfer Center of the Lodz University of Technology | Lodz University of Technology |
| Center for Technology Transfer of the Agricultural University of Hugo Kołłątaj in Krakow | University of Agriculture in Krakow |
| Technology Transfer Center of the Medical University of Warsaw | Medical University of Warsaw |
| Center for Technology Transfer and Innovation of the Kazimierz Wielki University in Bydgoszcz | Kazimierz Wielki University |
| Center for Technology Transfer and Enterprise Development | Applied Research Institute – Warsaw Institute of Technology LLC |
| Knowledge Transfer Center | Science and Technology Park of the Koszalin University of Technology |
| Knowledge Transfer Center | Medical University of Lublin |
| Center for Knowledge and Technology Transfer | Gdansk University of Technology |
| Center of Knowledge and Scientific and Technical Information. Center for Cooperation between Science and Economy | Wroclaw University of Science and Technology |
| Department of Innovation and Cooperation with the Economy—Technology Transfer Section | Lodz University of Technology |
| Department of Science and Technology Transfer of the Opole University of Technology | Opole University of Technology |
| Applied Research Institute of Warsaw University of Technology | Warsaw University of Technology |
| Institute of Innovation and Technology of the Białystok University of Technology | Bialystok University of Technology |
| Interdisciplinary Center of Modern Technologies of the Nicolaus Copernicus University in Toruń | The Nicolaus Copernicus University in Torun |
| National Center of Innovation and Technology Transfer | Higher School of Management and Coaching in Wroclaw |
| Lublin Development Foundation | Lublin Foundation for Development |
| Lublin Center for Technology Transfer | Lublin University of Technology |
| Center for Quality and Innovation | Faculty of Technical Sciences of the University of Warmia and Mazury in Olsztyn |
| Regional Center for Innovation and Technology Transfer | West Pomeranian University of Technology in Szczecin |
| Regional Innovation Center—Technology Transfer Center | UTP University of Science and Technology |
| Regional Center for Knowledge Transfer and Innovative Technologies at the State University of Applied Sciences in Nysa | University of Applied Sciences in Nysa |

**Table 1.** *Cont.*

| Name of the Transfer Technology Centre | Organization |
|---|---|
| SPIN-US | University of Silesia in Katowice |
| Independent Position for Innovation—BUSINESS POINT | Medical University of Gdansk |
| TechTransBalt | University of Gdansk |
| UNICO.AI Vojtěch Nosek | UNICO.AI |
| UWRC | University of Warsaw |
| University Office for Intellectual Property Protection and Technology Transfer | Medical University of Bialystok |
| University Center for Innovation and Technology Transfer of the University of Adam Mickiewicz | Adam Mickiewicz University in Poznan |
| University Center for Technology Transfer of the University of Warsaw | University of Warsaw |
| University Center for Technology Transfer | University of Rzeszow |
| Wrocław Technology Park | Wrocław Technology Park |
| Eastern Technology Transfer Center | University of Bialystok |
| Innovation and Technology Transfer Team, Poznań Science and Technology Park | Adam Mickiewicz University Foundation |
| Innovation, Technology and Analysis Team | Centre of Polymer and Carbon Materials PAN |
| All private organizations | |
| STB Innovation Center | STB Innovation Centre |
| Augere Capital Venture SK | Investment Fund |
| Development and Innovation Research Foundation | Foundation for Research, Development and Innovation |
| IBAkteria | Institute of Biotechnology and Antibiotics |
| International Eastern Innovation Center | Innovative Eastern Poland Association |
| Pre-incubator of Academic Entrepreneurship CTE/UR in Krakow | Ecotechnology Transfer Centre LLC |
| Radom Center of Innovation and Technology | Radom Center of Innovation and Technology |

Most of these centers are connected with universities and science institutions. There are three leaders with revenues of over a million USD and two strong followers. The rest of them are at the low-end [7]. Only eight TTCs are private organizations—see the second part of Table 1.

In Malaysia, however, the development of TTCs has been a bit slower than in Poland. Therefore, there are only seven centers nowadays. They are presented in Table 2.

**Table 2.** Transfer technology centers in Malaysia.

| Name of the Transfer Technology Centre | Organization |
|---|---|
| Universiti Malaya Centre of Innovation & Enterprise (UMCIE) | Universiti Malaya |
| Inovasi UKM (Centre of Innovation & Technology Transfer) | Universiti Kebangsaan Malaysia |
| Centre for Innovation and Consultation | Universiti Sains Malaysia |
| UTM Innovation and Commercialization Center (ICC) | Universiti Teknologi Malaysia |
| Putra Science Park | Universiti Putra Malaysia |
| Business Innovation & Commercialization Centre (BITCOM) | Universiti Teknologi MARA |
| Technology Exploitation & Delivery (TED) | Universiti Teknologi Petronas |

All of these technology transfer centers are run by the universities. Unlike in Poland, there are no private TTCs in Malaysia.

This overview of technology transfer centers presents the situation in the whole country. The research requires, however, more in-depth analysis. In Poland, one of the high-end technology transfer centers was created by Akademia Górniczo-Hutnicza (AGH) in Kraków. The goal of CTT AGH is to support the processes of commercialization and transfer of innovative technologies and knowledge. In connection with this, it cooperates with the scientific as well as the business community and organizations associating with entrepreneurs. There is a network of departmental brokers at the university, one at each faculty. Through this network, CTT AGH connects the needs of the industry with scientific teams as these specialists support SMEs with their knowledge and competencies in the complex meanders of the commercialization process of the research and development results.

In Malaysia, one of the interesting examples is the UiTM Business Innovation & Technology Commercialization Centre (BITCOM). It is the essential Technology Transfer Office dedicated to supporting the implementation of research effects. The goal of BITCOM is to manage the university's intellectual assets, which include innovations, inventions, and research achievements. BITCOM plays a critical role in incubating UiTM-based startups and spin-off companies by fostering cooperation between UiTM researchers and industry stakeholders. Above all, BITCOM reflects the belief that the convergence of innovation and academic research is a potent driver of positive change, ensuring that the benefits of university research improve society in meaningful ways.

Overall, Table 3 presents technology transfer and commercialization initiatives at AGH and UiTM from 2019 through 2023. There were 32 new commercialized projects at AGH in 2020. That makes the change at over 76% in comparison to 2019. UiTM had a total of 21 products commercialization in 2020, with two new items commercialized during the year. That makes the change at around 10%. The number of technology transfers at AGH in 2021 was quite high—38 projects. That makes the change at about 43%. The total number of new projects at UTiM climbed to 28 in 2021, with seven new product commercialization. The dynamics of technology transfers went up to over 33%. In 2022, the number of new projects at AGH dropped to 28 transfers. That makes the change at about 25%. However, at UiTM, there were 36 total product commercializations in 2022, including eight new products. That makes the change at 28.5%. In the next year, there was still a lower number of new transfers at AGH—23 projects. In connection with this, the dynamics of growth went down to 16.5%. UiTM had a total of 37 product commercialization by 2023, with one new product commercialized that year. Therefore, the growth rate decreased to around only 3%.

**Table 3.** Projects Commercialized and technology transfers at AGH and UiTM 2019-2023.

| AGH | | | | | |
|---|---|---|---|---|---|
| | **2019** | **2020** | **2021** | **2022** | **2023** |
| Cumulative Product Commercialization | 42 | 74 | 112 | 140 | 163 |
| New Product Commercialized | - | 32 | 38 | 28 | 23 |
| Dynamics of change (in %) | - | 76.1 | 43.2 | 25.0 | 16.5 |
| **UiTM** | | | | | |
| | **2019** | **2020** | **2021** | **2022** | **2023** |
| Cumulative Product Commercialization | 19 | 21 | 28 | 36 | 37 |
| New Product Commercialized | - | 2 | 7 | 8 | 1 |
| Dynamics of change (in %) | - | 10.5 | 33.3 | 28.5 | 2.7 |

Source: AGH, UiTM.

Clearly, both AGH and UiTM have been actively involved in commercializing concepts and goods over this time. However, the number of new projects, as well as dynamics of change, have been higher in the former university. Nevertheless, the data demonstrates the university's dedication to transforming research and ideas into practical products by highlighting a continual increase in cumulative product commercialization. The fluctuating number of new and total active items commercialized demonstrates the dynamic character of technology transfer operations, as well as their responsiveness to market demands and research outputs.

Commercialization initiatives, either at AGH or UiTM, are diversified and cross-industries, demonstrating a dedication to multidisciplinary innovation. The initiatives in the former university focus on IT, food technology, or engineering sectors. Unfortunately, the numbers of transfers broken down by sectors are not available. Most of the Polish companies did not agree to reveal the agreements with AGH. However, these data are available for UiTM—see Table 4. The concentration on engineering, healthcare, and medical technology shows a focus on fields of research with a large societal influence. Companies in education, finance, and management are present, indicating a realization of the necessity of commercialization in these disciplines as well.

**Table 4.** Number of Projects Commercialized by UiTM (by industry) in 2022.

| Industry | Number of Transfers |
|:---:|:---:|
| Engineering | 16 |
| Art & Design | 4 |
| System & Computer | 2 |
| Food Technology | 4 |
| Education | 1 |
| Finance & Management | 2 |
| Healthcate & Medical Technology | 7 |

Source: UiTM.

It is worth noting that the number of companies in each sector group might reveal insights into the university's strengths, areas of concentration, and prospective pathways for future research and technology transfer projects.

## 5. Discussion

The results obtained provide several valuable insights into the functioning of national technology transfer center systems, complementing research conducted so far in various countries across the globe, such as Singapore [18], China [19], Romania [20], United Kingdom [21] and Russia [22]. In particular, they provide a better understanding of the organizational and ownership structure of technology transfer systems [23], especially as a part of innovation [24] and entrepreneurial [25] ecosystem development. The results also provide important conclusions on the spatial distribution of national technology transfer systems [26–28].

Analysis of the changes in the projects' commercialization and technology transfers in both organizations under study aligns with the dynamic approach to developing technology-based relationships between industry and SME support actors [29,30]. This research also brings new knowledge to consider the scope of activities and industry-specific operating models of technology transfer centers [31–33]. This is particularly important in developing effective technology transfer mechanisms, methods, and practices [34,35], as well as building successful business models for technology transfer centers [36].

When discussing the results obtained, it should also be noted that technology transfer to industry is significantly affected by cultural factors [37], especially corporate culture [38]. In this case, the most beneficial is the corporate culture that includes innovation and risk-

taking, attention to detail, aggressiveness, stability, results orientation, people orientation, and team orientation [39]. This makes it possible to compile a list of follow-ups affecting the relative receptivity of both Malaysian and Polish corporate culture to technology transfer.

In Malaysia, corporate culture is characterized by such features as empowerment, team orientation, capability development, creating changes, customer focus, and organizational learning, which contribute significantly towards innovativeness [40]. Wang and Abdul-Rahman [41] found that the "monkey culture," which epitomizes teamwork and loyalty, is considered the most applicable type for the corporate culture in Malaysia. This is confirmed by Naqshbandi et al. [42] found that integrative culture dominates in Malaysian high-tech industries. This type of corporate culture pays equally high attention to employee development and harmony (internal integration) as well as customer orientation, social responsibility, and innovation (external adaptation).

According to Hofstede [43], the model Malaysian culture represents [44,45]: collectivism, high power distance, relatively high uncertainty avoidance (however, results are not consistent in this dimension), femininity (low masculinity) and relatively short-term orientation. At the same time, the specifics of this culture significantly and positively influence organizational learning, affective commitment [46], entrepreneurial innovativeness [47], and overall organizational performance [48]. In this context also, such Malaysian cultural dimensions as communication, reward, and recognition, as well as training and development, have a significant impact on organizational commitment [49]. Moreover, a study by Vasudevan et al. [50], confirm that corporate culture is an essential determinant of firm innovation and influences Malaysian SMEs' innovation management towards Industry 4.0. It strongly supports the relative receptivity of Malaysian corporate culture to technology transfer.

In Poland, corporate culture is characterized by such features as low context, achieved status, innovativeness, external focus, and task orientation [51]. However, in this study, businesses from Poland were characterized by moderate intensity of occurrence of all characteristics from all dimensions. Therefore, a clear classification of organizational culture to one of the types inside dimension is not proper in the case of all dimensions. According to Hofstede et al., [52], extended model Polish culture represents [53–55]: moderate individualism, high power distance, high uncertainty avoidance, rather masculinity, relatively short-term orientation (however results are not consistent in this dimension) and low indulgence.

Mazur and Zaborek [56] found that Polish culture positively influences an index of operational performance and ROI, according to Strychalska-Rudzewicz [57], the specifics of this culture also positively affect the innovation index of Poland; however, in the future, cultural changes should occur to give more power distribution to reduce power distance and to accept tolerance for change and ambiguity to reduce uncertainty avoidance. Zdunczyk and Blenkinsopp [58] detected the differentiation of corporate culture according to company ownership. Partly or fully foreign-owned companies operating in Poland appeared to be much more enabling of creativity and innovation than their wholly Polish-owned counterparts. Polish enterprises can benefit significantly from further effective assimilation of Western management philosophy and management methods.

In the perception of Polish managers from multicultural enterprises, the dominant types of corporate culture are adhocracy and the market [59]. With the adhocracy culture, the organization is dynamic, entrepreneurial, and creative. What ensures the organization's coherence is the willingness to take risks, experiment, and innovate. Emphasis is placed on maintaining a leading position in the field of new knowledge, products, or services, on readiness to change, and on facing new challenges. With the market culture, the organization is oriented much more strongly to external matters, to shaping its own position in the business environment than to internal matters. Enterprises with this type of culture focus primarily on developing relationships with other units to gain resources and competitive advantage. In other research, adhocracy and market culture were significantly

and positively associated with innovation [60]. It strongly supports the relative receptivity of Polish corporate culture to technology transfer.

Results obtained also contribute to a recent and forward-looking trend in the perception of technology transfer as a factor for sustainable development [61,62], which is especially important for developing countries [63]. Since science and technology have taken a prominent place in the United Nations framework of sustainable development goals [64] there is a search for a new, inclusive role for technology transfer in creating environmental- and societal-based innovations [65,66]. In this context, attention should also be paid to the economic prospects of technology transfer for sustainability.

These challenges are strongly emphasized, for example, in the technology transfer model oriented to sustainable development by Corsi et al. [67]. The model consists of 12 stages divided into four macro steps: (1) Plan, (2) Enable, (3) Implement, and (4) Evaluate, which range from identifying the technological need to documentation, registration, and feedback on implementation. In this proposal, there is a strong focus on the exploration and exploitation of economic opportunities and the generation of economic value. Much attention is also paid to investment and economic planning for technology transfer. According to the authors, economic issues are the most recurrent barriers to the transfer process, and it is important to address them. Barriers such as the high investment cost to obtain and implement the technology and the lack of subsidies or financial incentives to obtain innovative technologies can be supported. The authors propose the use of specific financial incentive mechanisms as an opportunity to promote the transfer and implementation of innovative and sustainable technologies, e.g., Global Environment Facility (GEF), Clean Development Mechanism (CDM); carbon credit for clean technologies, Official Development Assistance (ODA), Foreign Direct Investment (FDI) and International Financing Corporation (IFC). Another solution for supporting innovation and technology transfer is microfinancing (including competitively low-interest rate loans) [68,69], which is particularly valuable in the absence of commercial bank financing or other more traditional forms of entrepreneurial finance [70].

In addition, the proposed model pays significant attention to ensuring realistic contracts between the actors involved in the technology transfer process, according to Hidalgo and Albors [71], as well as Arenas and Gonzalez [72], the formal mechanisms may include license, university spin-off, contract of sale, delivery of technology to industry, technological alliance, production license, and marketing contract. The informal mechanisms are mainly knowledge capture and recruitment. This indicates the complexity of the relationships occurring in technology transfer processes. The presented research results also underline this complexity by showing the industry diversity of commercialized projects, which require the development of specific contracts that take into account the industry specificity of stakeholders in the technology transfer process.

## 6. Conclusions

In Poland, the main problem regarding the technology transfer is connected with the fact that this process is in the early phase of development. As there are not many TTCs, there is no need to create different regional networks. Nevertheless, the government of Poland supports the transfer of technology as it faces a significant challenge to meet the criteria for energy transformation imposed by the European Union. The efforts should be intensified as the results have been far from assumed targets. Therefore, the appropriate changes in regulations are expected, which will help to transfer technology from science institutions to SMEs.

However, the findings show that technology transfer and commercialization efforts both in Poland and Malaysia are on the right track. This demonstrates the universities' persistent dedication to turning research and innovative ideas into concrete products, as seen by the university's sustained growth in total product commercialization. It emphasizes the critical role that they play in promoting technological transfers, particularly for SMEs. Both AGH and UiTM should analyze and change their methods on a regular basis to

manage variations in new product commercialization and ensure the long-term viability active commercialized items.

The findings could be of interest to policymakers on macro and micro levels. The paper has some constraints, too. First of all, it presents detailed data for two technology transfer centers. Some information is available only for the Malaysian case. Therefore, it is not possible to undertake the statistical analysis. The lack of comparable macroeconomic data pertaining to SMEs in Malaysia and Poland is another of the study's limitations. Although technology transfer is the study's main focus, a more thorough economic analysis, especially one that addresses small and medium-sized businesses, would offer a wider viewpoint [73]. Furthermore, Hills et al. [74] acknowledge the possibility of investigating the convergence of marketing, management, and entrepreneurial practices between the European Union (EU) and Asia, where Poland and Malaysia are situated. Further investigation into transcontinental economic relationships is made possible by this avenue, which offers an exciting field for study. However, this paper lays the groundwork for subsequent research into the university's technology transfer programs, as well as their alignment with market dynamics and research outcomes.

**Author Contributions:** Conceptualization, formal analysis, supervision, M.W. and M.M.; writing—original draft preparation, collected data, data validation, performed the first data analyses, funding acquisition, All authors; the literature review, writing—original draft preparation, writing—review, M.W., M.M. and F.R.A. All authors have read and agreed to the published version of the manuscript.

**Funding:** This research was co-financed with funds from the Horizon Europe Framework Programme (HORIZON-MSCA-2021-SE-01-1)—Project Number: 101086381 and funds from the Polish Ministry of Education and Science within the framework of the program titled 'International Projects Co-Financed'—Project Number W 4/HE/2023 (Agreement Number: 5319/HE/2022/2023/2).

**Institutional Review Board Statement:** Not applicable.

**Informed Consent Statement:** Not applicable.

**Data Availability Statement:** The data presented in this study are available on request from the corresponding author.

**Conflicts of Interest:** The authors declare no conflict of interest.

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
