# Peer review of "Technology Transfer Centers as Support Instruments for SMEs—Comparative Analysis of Poland and Malaysia"

_sustainability, doi:10.3390/su152215814_

Round 1

Reviewer 1 Report

Comments and Suggestions for Authors

In the conclusion the following needs more work

quote

The findings could be of interest for policymakers on macro and microlevels. The 283 paper has some constraints, too. First of all, it presents the detailed data for two technol-284 ogy transfer centers. Some information is available only for the Malaysian case. Therefore, 285 it is not possible to undertake the statistical analysis. Nevertheless, this analysis lays the 286 groundwork for subsequent research into the university's technology transfer programs, 287 as well as their alignment with market dynamics and research outcomes.The findings could be of interest for policymakers on macro and microlevels. The 283 paper has some constraints, too. First of all, it presents the detailed data for two technol-284 ogy transfer centers. Some information is available only for the Malaysian case. Therefore, 285 it is not possible to undertake the statistical analysis. Nevertheless, this analysis lays the 286 groundwork for subsequent research into the university's technology transfer programs, 287 as well as their alignment with market dynamics and research outcomes.The findings could be of interest for policymakers on macro and microlevels. The 283 paper has some constraints, too. First of all, it presents the detailed data for two technol-284 ogy transfer centers. Some information is available only for the Malaysian case. Therefore, 285 it is not possible to undertake the statistical analysis. Nevertheless, this analysis lays the 286 groundwork for subsequent research into the university's technology transfer programs, 287 as well as their alignment with market dynamics and research outcomes.The findings could be of interest for policymakers on macro and microlevels. The 283 paper has some constraints, too. First of all, it presents the detailed data for two technol-284 ogy transfer centers. Some information is available only for the Malaysian case. Therefore, 285 it is not possible to undertake the statistical analysis. Nevertheless, this analysis lays the 286 groundwork for subsequent research into the university's technology transfer programs, 287 as well as their alignment with market dynamics and research outcomes.The findings could be of interest for policymakers on macro and microlevels. The 283 paper has some constraints, too. First of all, it presents the detailed data for two technol-284 ogy transfer centers. Some information is available only for the Malaysian case. Therefore, 285 it is not possible to undertake the statistical analysis. Nevertheless, this analysis lays the 286 groundwork for subsequent research into the university's technology transfer programs, 287 as well as their alignment with market dynamics and research outcomes.The findings could be of interest for policymakers on macro and microlevels. The 283 paper has some constraints, too. First of all, it presents the detailed data for two technol-284 ogy transfer centers. Some information is available only for the Malaysian case. Therefore, 285 it is not possible to undertake the statistical analysis. Nevertheless, this analysis lays the 286 groundwork for subsequent research into the university's technology transfer programs, 287 as well as their alignment with market dynamics and research outcomes.The findings could be of interest for policymakers on macro and microlevels. The 283 paper has some constraints, too. First of all, it presents the detailed data for two technol-284 ogy transfer centers. Some information is available only for the Malaysian case. Therefore, 285 it is not possible to undertake the statistical analysis. Nevertheless, this analysis lays the 286 groundwork for subsequent research into the university's technology transfer programs, 287 as well as their alignment with market dynamics and research outcomes.The findings could be of interest for policymakers on macro and microlevels. The 283 paper has some constraints, too. First of all, it presents the detailed data for two technol-284 ogy transfer centers. Some information is available only for the Malaysian case. Therefore, 285 it is not possible to undertake the statistical analysis. Nevertheless, this analysis lays the 286 groundwork for subsequent research into the university's technology transfer programs, 287 as well as their alignment with market dynamics and research outcomes.The findings could be of interest for policymakers on macro and microlevels. The 283 paper has some constraints, too. First of all, it presents the detailed data for two technol-284 ogy transfer centers. Some information is available only for the Malaysian case. Therefore, 285 it is not possible to undertake the statistical analysis. Nevertheless, this analysis lays the 286 groundwork for subsequent research into the university's technology transfer programs, 287 as well as their alignment with market dynamics and research outcomes.

end of quote

The only criticism I have of the paper is that there is not a point by point discussion as to the relative cultural factors leading to the formulation of the University Transfer in both countries. I.e. the topic is extremely important and it needs to be discussed, as that Malaysia and Poland have VERY different work cultures and emphasis as to what is, and is not important as to technology;. i.e. the authors should venture to speculate on this and do a laundry  list of follow ups as to what can be ascertained as to the relative receptivity of both Malaysian and Polish corporate culture to technology transfer. Aside from that the paper brings up many useful data points from a technology transfer case study stand point and I found it very well worth reading

From a policy stand point the overall paper results are outstanding and extremely important.

Author Response

Dear Sir/Madame,

We would like to thank for the valuable advice and comments on our manuscript. We appreciate them very much as they have improved our paper. We are also grateful to you for allowing us to revise our manuscript.

An analysis of the factors that confirm the receptivity of both Malaysian and Polish corporate culture to technology transfer has been conducted in the "Discussion" section.

Reviewer 2 Report

Comments and Suggestions for Authors

I strongly encourage the authors to include the Result and Discussion section to the manuscript rather than writing everything under the section “Technology transfer centers in Poland and Malaysia '' as a bulk content. I believe the separate content on the result and discussion sections could help readers to understand the major findings and author's viewpoint on it.

Author Response

Dear Sir/Madame,

We would like to thank for the valuable advice and comments on our manuscript. We appreciate them very much as they have improved our paper. We are also grateful to you for allowing us to revise our manuscript.

In the revised version, we expanded the structure of the paper by adding a "Discussion" section, where we analyze the results obtained in the context of previous studies and literature review.

Reviewer 3 Report

Comments and Suggestions for Authors

The manuscript has to undergone substantial improvements, prior to be accepted for publication at the Sustainability journal. Therefore, the following review comments can be supportive.

1. The Introduction section can be enhanced in a more solidified and extended citations’ overview.

2. In section 2. Literature review authors are recommended to somehow organize better how and to what extend 2 countries of large geographical distance to each other, at two different continents and with different entrepreneurial and managerial backgrounds could be considered as convergent and cooperative to each other. What more, sustainability prospects should be solely as “economic”, not as “social”, or “environmental”, or “energy”, or “technological”.  These issues have to be discussed in the Discussion section.

3. Based on the aforementioned review comment, another issue of consideration is section 2 to contain a separate introduction of the a) Poland, b) Malaysia economic status during the last 2-3 decades of overview, as well as a third subsection c) to coordinate the financial, marketing, business, context of comparing their analysis. Besides, the aforementioned limitations of: distance, different cultures, different technological and energy situations and priorities concurring in both countries.

4. In the Methodology authors are recommended to specify what are the common ground issues, business sectors, technological innovations, under which the co-presentation of the SMEs between the two countries takes place. The exact calendar year of collecting, or issuing, the input data of this study can be specified.

5. The roles of microfinancing with competitively low interest rate loans, as well as the realistically signed trading contracts between specific SMEs-sectors, could be also denoted in the Discussion section.

6. The narrative flow and the existing citations’ number like more as a “report”, or “short communication” study, not as an “article”. This suitable characterization of the manuscript is an issue to be discussed with the debuted Sustainability editor who is responsible for this study submitted.

7. In Tables 1 and 2 one extra column has to be added, to each of them, clearly stating what are the types of technological evaluation took place? Besides, if there is “organizational”, “technological”, convergence among these entries, then, a third Table has to be formulated co-presenting both countries, under the same domains of “organization”, “technology”, referring to.

8. The citations have been presented in mixed way: per authors’ name in text and per number in the References section. A consistency between these two different forms is needed here.

9. Another limitation of the study is that there are not economics-marketing-management macroeconomic data referring to the SMEs per each one country, Poland and Malaysia. Even more, it would be interesting if authors, beyond the  Poland and Malaysia countries, could be a marketing-managerial-entrepreneurial convergence between the two continents, EU and Asia, in which these two countries are positioned? 2-3 sentences are adequate.

10. The concluding remarks can also contain a common ground of “sustainability” orientation, since it is not straightforward or directly/automatically perceived if the technology is directly related to “(a “general”) sustainability between SMEs” at two different contexts? This critical statement has to be approached in a creative, scientific specialization, way by authors. 

Author Response

Dear Sir/Madame,

We would like to thank for the valuable advice and comments on our manuscript. We appreciate them very much as they have improved our paper. We are also grateful to you for allowing us to revise our manuscript. Let us respond to your comments in the following points:

  1. We enhanced the Introduction by additional information.
  2. We changed the organization of the section 2. Literature review. Moreover, economic prospects of technology transfer for sustainability have been discussed in the “Discussion” section.
  3. We included the seperate subsections regarding Poland, Malaysia, and both countries. Furthermore, we implemented information about economic status in the last 25 years either for Poland and Malaysia.
  4. We specified the common ground issues for the analyzed countries. Moreover, we delivered information about the calendar of collecting the data.
  5. The roles of microfinancing as well as the realistically signed trading contracts between technology transfer actors have been denoted in the “Discussion” section.
  6. The new structure of the paper as well as references are considered as  typical for a research apaper.
  7. The issues regarding cinvergence among entries are implemented in the "Discussion" section.
  8. We deleted the numbers of publication in the Reference section. It is done according ot the alphabetical order.
  9. Additional infomation aboutht the cinvergence in Europe and Asia awere added to the section Conclusion.
  10. The remarks about a common ground of "sustainability" orientation are delivered in the new section "Discussion".

Round 2

Reviewer 3 Report

Comments and Suggestions for Authors

The manuscript has been satisfactorily revised and creatively improved, comparing to the initial one. In this respect, it sustains novel features of technological, managerial, and widely environmental interest, thus, it can be accepted for publication at the Sustainability journal as is.